# SpeechGPT: Empowering Large Language Models with Intrinsic Cross-Modal Conversational Abilities

**Dong Zhang, Shimin Li, Xin Zhang, Jun Zhan, Pengyu Wang,**
**Yaqian Zhou,* Xipeng Qiu***
School of Computer Science, Fudan University
Shanghai Key Laboratory of Intelligent Information Processing, Fudan University
{dongzhang22,xin_zhang22,jzhan22,pywang22}@m.fudan.edu.cn
{smli20,zhouyaqian,xpqiu}@fudan.edu.cn

## Abstract

Multi-modal large language models are regarded as a crucial step towards Artificial General Intelligence (AGI) and have garnered significant interest with the emergence of ChatGPT. However, current speech-language models typically adopt the cascade paradigm, preventing inter-modal knowledge transfer. In this paper, we propose SpeechGPT, a large language model with intrinsic cross-modal conversational abilities, capable of perceiving and generating multi-modal content. With discrete speech representations, we construct SpeechInstruct, the first large-scale cross-modal speech instruction dataset. Additionally, we employ a three-stage training strategy that includes modality-adaptation pre-training, cross-modal instruction fine-tuning, and chain-of-modality instruction fine-tuning. The experimental results demonstrate that SpeechGPT has an impressive capacity to follow cross-modal human instructions and highlight the potential of handling multiple modalities with one model. Code and models are available in https://github.com/0nutation/SpeechGPT. Demos are shown in https://0nutation.github.io/SpeechGPT.github.io/.

## 1 Introduction

Large language models (OpenAI, 2023; Touvron et al., 2023) have performed astonishingly on various natural language processing tasks. Meanwhile, multi-modal large language models, such as GPT-4, PALM-E (Driess et al., 2023), and LLaVA (Liu et al., 2023), have explored the ability of LLMs to understand multi-modal information. However, a significant gap exists between current LLMs and general artificial intelligence (AGI). First, most current LLMs can only perceive and understand multi-modal content but cannot spontaneously generate multi-modal content. Second, continuous signals

*Corresponding author

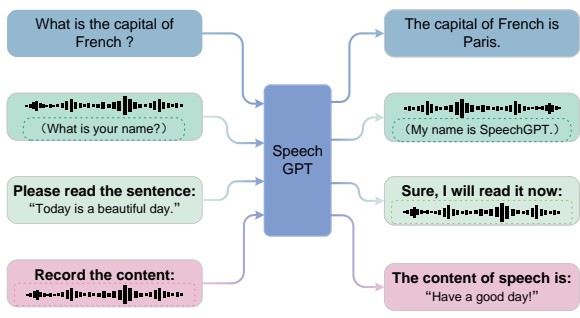

Figure 1: SpeechGPT's capabilities to tackle multiple cross-modal tasks.

like images and speech cannot be adapted directly to LLMs that receive discrete tokens.

The current speech-language model mainly adopts a cascading paradigm (Huang et al., 2023a) i.e., the LLM is connected with an automatic speech recognition (ASR) model or a text-to-speech (TTS) model in tandem, or the LLM is employed as a control hub, with several speech processing models (Cheng et al., 2023a,b,c) are integrated to cover multiple audio or speech tasks (Huang et al., 2023a; Shen et al., 2023). Some prior work on generative spoken language models involves encoding the speech signal into a discrete representation (Baevski et al., 2020; Hsu et al., 2021; Zhang et al., 2023a) and modeling it with language models (Lakhotia et al., 2021; Borsos et al., 2022; Zhang et al., 2023d; Wang et al., 2023; Zhang et al., 2023c).

While capable of perceiving and generating speech, the existing cascaded methods or spoken language models still have several limitations. First, the LLM in the cascaded model only functions as a content generator. Since the representations of speech and text are not aligned, the LLM's knowledge cannot be transferred to the speech modality. Second, the cascade approach (Shen et al., 2023; Huang et al., 2023a) suffers from the loss of paralinguistic signals such as emotion and prosody. Third, existing spoken language models (Wang

et al., 2023; Zhang et al., 2023d) only synthesize speech but fail to comprehend its semantic information, preventing them from achieving true cross-modal perception and generation.

In this paper, we propose SpeechGPT, a large language model with intrinsic cross-modal conversational abilities, capable of perceiving and generating multi-modal content. We perform speech discretization with a self-supervised trained speech model to unify the modality between speech and text. The discrete speech tokens are then expanded into the vocabulary of the LLM, thus endowing the model with an inherent competence to perceive and generate the speech.

To provide the model with the capacity to handle multi-modal instructions, we build the first speech-text cross-modal instruction-following dataset SpeechInstruct. Specifically, we discretize the speech to discrete units (Hsu et al., 2021) and construct the cross-modal unit-text pair based on the existing ASR dataset. Meanwhile, we construct hundreds of instructions for diverse tasks with GPT-4 to simulate actual user instructions as illustrated in Appendix B. In addition, to further enhance the model's cross-modal capability, we designed the Chain-of-Modality instruction data, i.e., the model receives the speech command, thinks about the process in text, and then outputs the response in speech.

For better cross-modal transfer and efficient training, SpeechGPT undergoes a three-stage training process: modality-adaptation pre-training, cross-modal instruction fine-tuning, and chain-of-modality instruction fine-tuning. The first stage enables speech comprehension for SpeechGPT with the discrete speech unit continuation task. The second stage employs the SpeechInstruct to improve the model's cross-modal capabilities. The third stage utilizes parameter-efficient LoRA (Hu et al., 2021) fine-tuning for further modality alignment.

To evaluate the effectiveness of SpeechGPT, we conduct a wide range of human evaluations and case analyses to estimate the performance of SpeechGPT on textual tasks, speech-text cross-modal tasks, and spoken dialogue tasks. The results demonstrate that SpeechGPT exhibits a strong ability for unimodal and cross-modal instruction following tasks.

Our contributions include the following:

- We build the first multi-modal large language model that can perceive and generate multi-modal contents.
- We construct and release SpeechInstruct, the first large-scale speech-text cross-modal instruction-following dataset.
- We build the first spoken dialogue LLM with strong human instruction following ability and spoken dialogue ability.
- We show great potential to incorporate other modalities into LLMs through discrete representations.

## 2 Related Work

**Multi-modal Large Language Model** Current multi-modal LLMs predominantly focus on the visual domain, feeding continuous representations obtained from pre-trained visual encoders into LLMs, facilitating full-parameter or parameter-efficient training on visual-language data (OpenAI, 2023; Huang et al., 2023b; Zhang et al., 2023b). Palm-E (Driess et al., 2023) integrates the 540B PaLM (Chowdhery et al., 2022) and 22B Vision Transformer (Dosovitskiy et al., 2021) into the largest vision-language model. LLaVA (Liu et al., 2023) leverages pre-trained CLIP (Radford et al., 2021) visual encoder and LLaMA (Touvron et al., 2023) and conduct instruct tuning on GPT4-assisted visual instruction data. X-LLM (Chen et al., 2023) converts multi-modalities into representations with X2L interfaces as the inputs of the large language model. However, such structures only enable LLMs to process multi-modal input, without ability to generate multi-modal output. Diverging from prior studies, our approach emphasizes the development of a speech-centric multi-modal LLM, endowing it with the proficiency to accommodate both multi-modal input and output.

**Generative Spoken Language Model** Discrete self-supervised representation based spoken generative language modeling is making remarkable progress on large-scale speech dataset training (Nguyen et al., 2022). AudioLM (Borsos et al., 2022) proposes to model speech based on audio codecs together with semantic codes, which can synthesize speech in a textlesss setting. VALL-E (Wang et al., 2023) builds a generative spoken language model on audio codecs and treat Text-to-Speech as a conditional generation task. However, these models are designed for a specific task and failed to benefit from LLMs. SpeechGPT is built upon the foundation of LLM and transfers LLM's knowledge to speech modality, con-

sequently obtaining better task generalization and human-instruction following ability.

**Speech-Enabled LLM Interaction** Following the emergence of ChatGPT, several studies have concentrated on the integration of expert speech models with LLMs to enable direct speech interaction with LLMs. HuggingGPT (Shen et al., 2023) facilitates task decomposition of human instructions by LLMs and allows the invocation of models from Huggingface to accomplish specific tasks, encompassing a range of automatic speech recognition (ASR) and text-to-speech models. Audio-GPT (Huang et al., 2023a) leverages a variety of audio foundation models to process complex audio information and connect LLMs with input/output interface (ASR, TTS) for speech conversations. However, these models exhibit increased complexity, demand extensive resources, and are prone to the unavoidable error accumulation problems. Our approach enables speech interaction with LLMs without relying on ASR or TTS systems, circumventing the aforementioned drawbacks.

# 3 SpeechInstruct Construction

Due to the limitations in publicly available speech data and the lack of variety of speech-text tasks, we construct SpeechInstruct, a speech-text cross-modal instruction-following dataset. This dataset consists of two parts, the first part is called Cross-Modal Instruction, and the second part is called Chain-of-Modality Instruction. The construction process of SpeechInstruct is illustrated in Figure 2.

## 3.1 Cross-modal Instruction

**Data Collection** We collect several large-scale English ASR datasets to construct Cross-Modal Instruction, including Gigaspeech (Chen et al., 2021), Common Voice (Ardila et al., 2020), and LibriSpeech (Panayotov et al., 2015). We employ mHuBERT[1] as the speech tokenizer to discretize speech data into discrete units and remove the repetitive units of adjacent frames to get reduced units. Ultimately, we obtain 9 million unit-text data pairs.

**Task Description Generation** We generate ASR and TTS task descriptions that are compatible with speech-text data pairs. Unlike the Self-Instruct method (Wang et al., 2022), we generate descriptions through a zero-shot approach. Specifically, we directly input the prompts shown in Appendix A

into OpenAI GPT-4 to generate task descriptions. Our generation method yields 100 instructions for each task and some examples are shown in Appendix B.

**Instruction Formatting** For a discrete unit sequence $U$ and its associated transcription $T$, we determine whether it will be used for constructing an ASR task or a TTS task based on the probability $p$. Subsequently, we randomly select a description $D$ from the corresponding task description. This results in a triplet consisting of the task description, discrete unit sequence, and transcription, denoted as $(D, U, T)$. Following this, the triplet is assembled into an instruction using the template: **[Human]:**$\{D\}$**. This is input:** $\{U\}$**<eoh>.[SpeechGPT]:** $\{T\}$**<eos>.**.

## 3.2 Chain-of-Modality Instruction

**Speech Instruction Generation** Due to the lack of instruction data with speech input and speech output, we trained a text-to-unit generator to convert text instruction data into speech instruction data. Specifically, the text-to-unit generator adopts a Transformer encoder-decoder architecture. We trained it on LibriSpeech unit-text pairs in Cross-modal Instruction. We select 37,969 samples from the moss-002-sft-data dataset [2] whose response length is shorter than 35 words. And we convert both their instructions and responses into unit sequences through the text-to-unit generator. As a result, we obtained 37,969 quadruplets composed of speech instructions, text instructions, text responses, and speech responses, denoted as $(SpeechI, TextI, TextR, SpeechR)$.

**Instruction Formatting** Using the above quadruplets, we could construct chain-of-thought style instructions for four input-output formats, namely Speech Instruction-Speech Response, Speech Instruction-Text Response, Text Instruction-Speech Response, and Text Instruction-Text Response. Their corresponding templates can be found in Appendix C.

## 3.3 SpeechInstruct Evaluation Set

We constructed cross-modal dialogue datasets under different scenarios to evaluate whether SpeechGPT could take on various roles. Specifically, these included a talking encyclopedia, personal assistant, chat partner, poet, psychologist,

---

[1]https://dl.fbaipublicfiles.com/hubert/mhubert_base_vp_en_es_fr_it3.pt

[2]https://huggingface.co/datasets/fnlp/moss-002-sft-data

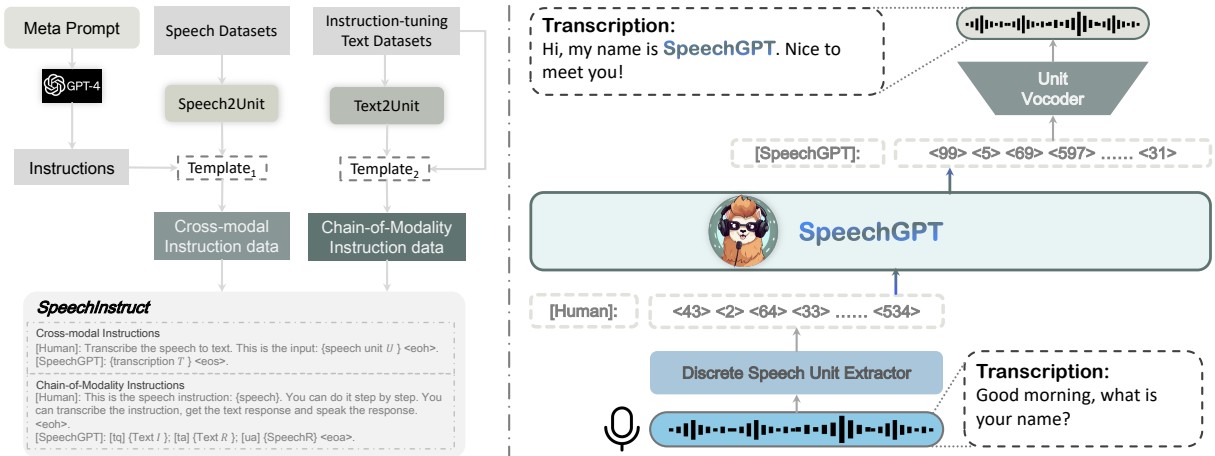

Figure 2: **Left**: An overview of SpeechInstruct construction process. The SpeechInstruct dataset consists of two parts: Cross-modal Instruction data and Chain-of-Modality Instruction data. $Template_1$ is shown in 3.1. $Template_2$ is shown in Appendix C. **Right**: An illustration of SpeechGPT model structure.

and educational assistant. For each role, we provide 10 manually authored instruction-response pairs written by ourselves. We use a pre-trained text-to-speech model [3] to convert the text into corresponding speech. We then employ mHuBERT to discretize speech data into discrete units as described in Section 3.1. Ultimately, for each role, we obtained 10 quadruplets composed of speech instructions, text instructions, text responses, and speech responses.

## 4 SpeechGPT

### 4.1 Model Structure

A unified framework is designed to provide architecture compatibility across different modalities. As shown in Figure 2, our model consists of three main components: *discrete unit extractor*, *large language modal* and *unit vocoder*. Under this architecture, LLM can perceive multi-modal inputs and generate multi-modal outputs.

**Discrete Unit Extractor** The discrete unit extractor utilizes the Hidden-unit BERT (HuBERT) model (Hsu et al., 2021) to transform continuous speech signals into a sequence of discrete units, . HuBERT is a self-supervised model that learns by predicting discrete labels for masked audio segments based on k-means clustering applied to the model's intermediate representations. It features a combination of 1-D convolutional layers and a Transformer encoder to encode speech into continuous intermediate representations, with a k-

means model further converting these representations into a sequence of cluster indices. Subsequently, adjacent duplicate indices are removed, resulting in a discrete units sequence represented as $U = (u_1, u_2, \ldots, u_T)$, $u_i \in 0, 1, \ldots, K-1$, $\forall 1 \le i \le T$, with $K$ denoting the total number of clusters.

**Large Language Model** We employ the Meta AI LLaMA (Touvron et al., 2023) model as our Large Language Model. LLaMA comprises an embedding layer, multiple transformer blocks, and an LM head layer. The total number of parameters in LLaMA ranges from 7B to 65B. Drawing from an extensive training dataset of 1.0 trillion tokens, LLaMA demonstrates competitive performance compared to the substantially larger 175B GPT-3 across various NLP benchmarks.

**Unit Vocoder** Due to limition of single speaker unit vocoder in (Polyak et al., 2021), we train a multi-speaker unit HiFi-GAN to decode the speech signal from the discrete representation. The HiFi-GAN architecture consists of a generator **G** and multiple discriminators **D**. The generator uses look-up tables (LUT) to embed discrete representations and the embedding sequences are up-sampled by a series of blocks composed of transposed convolution and a residual block with dilated layers. The speaker embedding is concatenated to each frame in the up-sampled sequence. The discriminator features a Multi-Period Discriminator (MPD) and a Multi-Scale Discriminator (MSD), which have the same architecture as (Polyak et al., 2021).

---

[3] https://huggingface.co/facebook/fastspeech2-en-ljspeech

## 4.2 Training

To incorporate speech discrete representation into LLM, we expand the vocabulary and corresponding embedding matrix first. We divide the training process into three stages. The first stage is Modality-Adaptation Pre-training on unpaired speech data. The second stage is Cross-modal Instruction Fine-Tuning. The third stage is Chain-of-Modality Instruction Fine-Tuning.

**Expanding Vocabulary** Given original LLM vocabulary $V$ of size $|V|$, to integrate speech discrete representations into LLM, we expand the vocabulary with an additional set of unit tokens $V'$, of size $|V'| = K$. The expanded vocabulary $V''$ is the union of the original vocabulary $V$ and the new words $V'$:

$$V'' = V \cup V' \quad (1)$$

We denote the original word embedding matrix as $E \in \mathbb{R}^{|V| \times d}$, where $d$ is the dimension of word embeddings. To accommodate the expanded vocabulary, we need to create a randomly initialized word embedding matrix $E' \in \mathbb{R}^{|V''| \times d}$. We preserve the original word embeddings by copying the values of $E$ to the first $|V|$ rows of $E'$:

$$E'[0:|V|,:] = E \quad (2)$$

Finally, we replace the original vocabulary and word embedding matrix with the new vocabulary $V''$ and the word embedding matrix $E'$.

**Stage 1: Modality-Adaptation Pre-training** To enable LLM to handle discrete units modality, we utilize an unlabeled speech corpus to train LLM in a next-token prediction task. This approach aligns with the text pre-training objective of LLM. Given unlabeled speech corpus $C$ consisting of speech $U_1, U_2, \ldots, U_m$ and LLM denoted as $L_1$, the negative log-likelihood loss can be formulated as:

$$\mathcal{L}(L|C) = -\sum_{j=1}^{m} \sum_{i=1}^{n_j} \log P(u_{i,j}|u_{<i,j}; L) \quad (3)$$

where $m$ is the number of speech in dataset $C$, $n_j$ is the number of discrete unit token in speech $U_j$, and $u_{i,j}$ represents the i-th unit token in the j-th speech.

**Stage 2: Cross-modal Instruction Fine-Tuning** In this stage, we align speech and text modalities utilizing paired data. We mix Cross-modal Instruction in SpeechInstruct with moss-002-sft dataset to derive mix dataset $I$, which consists of samples $T_1, T_2, \ldots, T_x$. We fine-tune the model $L$ obtained from the first stage on $I$.

Each sample $T_j$ consisting of $t_1, t_2, \ldots, t_{n_j}$ is formed by concatenating a prefix and a text. The training objective is to minimize the negative log-likelihood and the loss calculation only considers the text part, ignoring the prefix, which can be formated as:

$$\mathcal{L}(L|I) = -\sum_{j=1}^{x} \sum_{i=p_j+1}^{y_j} \log P(t_{i,j}|t_{<i,j}; L) \quad (4)$$

where $x$ is the number of samples in corpus $I$, $y_j$ is the total number of tokens in sample $T_j$, $p_j$ is the number of tokens in the prefix part of $T_j$, and $t_{i,j}$ represents the i-th word in $T_j$.

**Stage 3: Chain-of-Modality Instruction Fine-Tuning** After obtaining the model in stage 2, we utilizes parameter-efficient Low-Rank Adaptation (LoRA) (Hu et al., 2021) to fine-tune it on Chain-of-Modality Instruction in SpeechInstruct. We add LoRA weights (adapters) to the attention mechanisms and train the newly added LoRA parameters. We adopt the same loss function as stage 2.

## 5 Experiments

### 5.1 Experimental Setups

**Datasets** For modality-adaption pre-training, we use LibriLight (Kahn et al., 2020) which contains 60K hours of unlabelled English audiobook speech. For cross-modal instruction fine-tuning stage, we use Gigaspeech (Chen et al., 2021), Common voice (Ardila et al., 2020) and LibriSpeech (Panayotov et al., 2015) dataset and moss-002-sft-data dataset, which is illustrated in detail in 3.1. For chain-of-modality instruction fine-tuning stage, we use moss-002-sft-data dataset, which is illustrated in detail in 3.2.

**Configuration** We employ LLaMA-13B (Touvron et al., 2023) as our backbone model for a trade-off between performance and computational resources available. For stage 1, we use 96 A100 GPUs and train for 900 steps with batch size 768. For stage 2, we use 96 A100 GPUs and train for 2100 steps with batch size 1536. For stage 3, we use 8 A100 GPUs and train for 4200 steps with batch size 128.

Details about training hyperparameters are shown in Appendix D. For decoding, we set the maximum sequence length to 2048 and set the temperature to 0.8. We use Top-$k$ sampling with $k$=60. We also use Top-$p$ sampling with p=0.8.

## 5.2 Baselines

We establish two cascaded cross-modal conversational systems as our baselines. The first model, referred to as *Speech-Alpaca-13B*, consists of an off-the-shell ASR system [4], Alpaca 13B (Taori et al., 2023) as well as a pre-trained TTS system [5]. The second model, named *Speech-LLaMA-MOSS-002*, incorporates the same ASR and TTS system, along with a large language model obtained by performing supervised fine-tuning on LLaMA-13B using MOSS-sft-002 as the training dataset.

## 5.3 Evaluation

We evaluate the cross-modal instruction-following capabilities of SpeechGPT across four tasks: *speech-to-speech instruction-following* (S2SIF), *speech-to-text instruction-following* (S2TIF), *text-to-speech instruction-following* (T2SIF), and *text-to-text instruction-following* (T2TIF).

**Data** We randomly select 40 samples from the AlpacaEval dataset [6] and use the pre-trained TTS model in Section 3.3 to convert the text into corresponding speech. We then employ mHuBERT to discretize speech data into discrete units as described in Section 3.1. These are combined with the SpeechInstruct Evaluation Set to constitute our test set, which contains 100 samples. Each sample is a quadruplet composed of a speech instruction, text instruction, text response, and speech response. We denote them as ground truth.

**ChatGPT Score** We utilize ChatGPT (GPT-3.5-turbo) to assess the cross-modal instruction-following performance. For tasks that include speech, we leveraged the pre-trained ASR model in section 5.2 to transform the speech into its corresponding text, which is subsequently submitted for evaluation. Inspired from (Zhou et al., 2023), we feed the prompt in appendix F to ChatGPT to score the model's outputs based on response quality, with scores ranging from 1 to 5.

---

[4] https://huggingface.co/facebook/wav2vec2-large-960h-lv60-self
[5] https://huggingface.co/facebook/fastspeech2-en-ljspeech
[6] https://github.com/tatsu-lab/alpaca_eval

**Human Opinion Score** Following (Nguyen et al., 2022), we calculate the human opinion score of the generated examples through crowdsourcing. These opinions are based on two dimensions: the content mean opinion score (CMOS) for content and meaningfulness quality, and the naturalness mean opinion score (NMOS) for speech naturalness and fluency. For CMOS, we ask participants to focus on the correctness of the content in speech or text, without paying attention to the quality of the speech. For NMOS, we direct participants to focus on the quality, smoothness, and naturalness of the speech, without considering its content. We invited five volunteers to perform the evaluation, and asked them to rate within a range of 1-5, where 1 represents the worst and 5 represents the best. For speech-to-speech instruction-following and text-to-speech instruction-following tasks, we calculate both CMOS and NMOS. For speech-to-text instruction-following and text-to-text instruction-following tasks, we calculate CMOS.

## 5.4 Main Results

**Content** As shown in Table 1, taking into account the comprehensive evaluation of ChatGPT Score and CMOS, SpeechGPT demonstrates superior performance on speech instructions (S2SIF and S2TIF) compared to the two baseline systems. This indicates that SpeechGPT outperforms the ASR model in the cascaded system when it comes to understanding speech content. From the perspective of CMOS, SpeechGPT achieves performance similar to the baseline systems on T2SIF and T2TIF tasks, indicating that SpeechGPT still possesses commendable text and speech generation capabilities. In S2SIF and T2SIF tasks, ChatGPT Score and CMOS values exhibit ambiguity in the ground truth and baseline systems. This can be attributed to speech responses being synthesized by TTS system, which can have errors in pauses between sentences. This introduces significant errors for longer responses, leading to incorrect text after being processed by the ASR system, thereby reducing the ChatGPT score. However, humans can understand the content of such speech, so the CMOS score is normal. Cases of cross-modal instruction-following can be found in Appendix G.

**Speech Quality** As shown in Table 1, SpeechGPT exhibits significantly higher NMOS values compared to the baseline systems. This indicates that the speech responses generated by SpeechGPT out-

| Methods | ChatGPT Score | | | | Human Opinion Score | | | | | | | |
|---|---|---|---|---|---|---|---|---|---|---|---|---|
| | | | | | CMOS | | | | NMOS | | | |
| | S2SIF | S2TIF | T2SIF | T2TIF | S2SIF | S2TIF | T2SIF | T2TIF | S2SIF | S2TIF | T2SIF | T2TIF |
| Ground Truth | 2.85* | 3.74 | 2.91* | 3.93 | 3.78 | 3.89 | 3.95 | 4.12 | 3.18 | - | 3.20 | - |
| *Baselines: cascaded cross-modal conversational systems* | | | | | | | | | | | | |
| Speech-Alpaca-13B | 2.74 | 3.31 | 2.71 | 3.83 | 3.39 | 3.42 | 3.71 | 3.75 | 3.12 | - | 3.13 | - |
| Speech-LLaMA-MOSS-002 | 2.87 | 3.50 | 3.23 | **3.82** | 3.38 | 3.44 | **3.74** | **3.83** | 3.14 | - | 3.11 | - |
| SpeechGPT | **3.42** | **3.52** | **3.53** | 3.64 | **3.42** | **3.49** | 3.57 | 3.69 | **3.65** | - | **3.62** | - |

Table 1: Main Results of SpeechGPT. S2SIF refers to speech-to-speech instruction-following, S2TIF is speech-to-text instruction-following, T2SIF denotes text-to-speech instruction-following and T2TIF represents text-to-text instruction-following. ChatGPT score is obtained through ChatGPT evaluatation. CMOS refers to content mean opinion score. NMOS denotes naturalness mean opinion score. *: The low ChatGPT Score for speech responses in Ground Truth is due to them being synthesized by TTS system, which can have errors in pauses between sentences. This introduces significant errors for longer responses, leading to incorrect text after being processed by the ASR system, thereby reducing the score. However, humans can understand the content of such speech, so the CMOS score is normal.

| Training | Inference | ChatGPT Score |
|---|---|---|
| Standard | Standard | 2.15 |
| Standard | CoM | 2.12 |
| CoM | Standard | 2.35 |
| CoM | CoM | **3.42** |

Table 2: ChatGPT Score on speech-to-speech instruction-following task. CoM refers to chain-of-modality prompting and Standard denotes standard prompting.

perform the TTS system in the cascaded system in terms of audio quality and prosody. More detailed speech prosody analysis are located in Section **??**.

## 6 Analysis

### 6.1 Chain-of-modality prompting matters

Table 2 shows ChatGPT Scores on speech-to-speech instruction-following task for models utilizing standard prompting and chain-of-modality prompting during training and inference stages respectively. Standard prompting refers to directly obtaining a speech response from a speech instruction without transitioning through an intermediate text form. The template can be located in Appendix E. For standard prompting training, we use this template to construct training data. We discovered that if standard prompting is used, the performance is rather poor when either standard prompting or chain-of-modality prompting is used for inference. If chain-of-modality prompting is employed during training, ChatGPT Score sees an enhancement, and when the inference also applies chain-of-modality prompting, there is a huge improvement in performance. This indi-

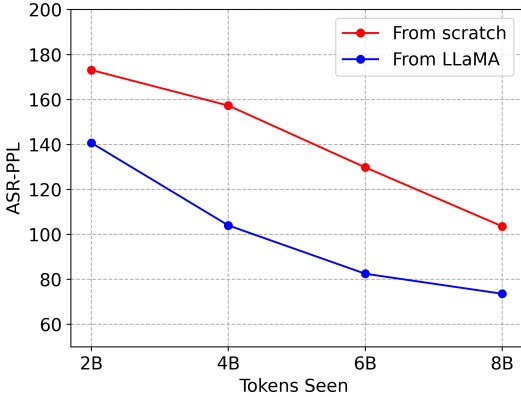

Figure 3: ASR-PPL of speech continue task on 100 utterances from LibriSpeech test-clean set. *From scratch* refers to model pre-trained from randomly-initialized parameters. *From LLaMA* denotes model pre-trained from LLaMA.

cates that **chain-of-modality prompting matters in both training and inference**. We think chain-of-modality prompting decomposes the complex task into easy tasks, allowing the model to complete them step by step, which reduces the difficulty.

### 6.2 Can text knowledge benefit speech modality?

SpeechGPT originates from a text pre-trained model, LLaMA. Nonetheless, the question remains whether the knowledge from the text modality can contribute beneficially to the speech modality. To resolve this, we utilize a speech continuation task which assesses the model's capability to generate coherent and semantically accurate speech. We compare the performances of two models on this task: one model is pre-trained from LLaMA, while

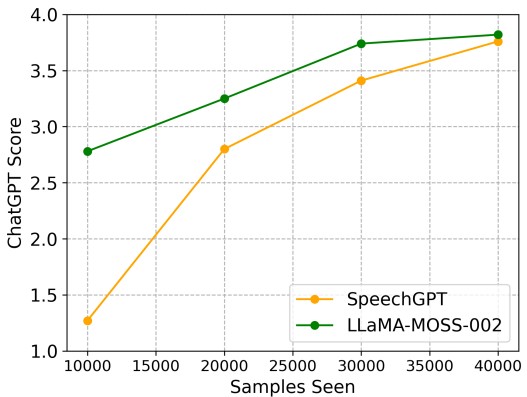

Figure 4: ChatGPT Score on text-to-text instruction-following task. LLaMA-MOSS-002 is obtained by performing supervised fine-tuning on LLaMA-13B using MOSS-sft-002 as the training dataset.

the other model is trained from scratch.

We utilize LibriSpeech test-clean set for evaluation, where we randomly select 100 utterances, and use the first 3 seconds of each utterance as a prompt. The 3-second speech prompt is converted into discrete units by mHuBERT. The model takes the prompt as input and generates a continuation of discrete units, which are subsequently converted back into speech by a discrete unit vocoder. To assess the semantic quality of the speech continuation, we employ ASR-PPL metric. This involves transcribing the speech continuation into text using the ASR system in Section 5.2 and calculating the perplexity of the transcripts using GPT-3.5 text-devinci-003 model. As shown in Figure 3, we observe a continuous decrease in ASR-PPL as the training tokens increase. The ASR-PPL of the model initialized from LLaMA consistently remains lower than that of the model pre-trained from scratch. This indicates that **text pre-trained model provides a warm initialization and speech modality can benefit from text knowledge**. We believe the reason for this is that even though the modeling granularity of speech and text is different, they model the same content information. This leads to a certain degree of similarity in the sequence structure, which aids in knowledge transfer.

### 6.3 Does SpeechGPT Sacrifice Text Capability as a Trade-off?

Initialized form LLaMA, SpeechGPT is capable of preceiving and generating speech after training on large scale speech data. However, does SpeechGPT sacrifice text capability as a trade-off? To draw conclusions, we compared the text-to-text

instruction-following ability of SpeechGPT with LLaMA-MOSS-002. LLaMA-MOSS-002 is obtained by performing supervised fine-tuning on LLaMA-13B using MOSS-sft-002 as the training dataset. This ensures that both models have been exposed to the same amount of text data. We evaluated both models using the test set from Section 5.3.

As depicted in Figure 4, with an increase in training samples, both LLaMA-MOSS-002 and SpeechGPT's ChatGPT Score gradually improve. Although SpeechGPT consistently remains lower than LLaMA-MOSS-002. the performance gap between them gradually decreases. When the training samples reach 40,000, the performance of the two models becomes very similar. This suggests that **SpeechGPT still retains text capability**. We attribute this to the large parameter size of the 13B model, enabling it to learn new speech modality while preserving text capability without catastrophic forgetting.

## 7 Conclusion

This work presents SpeechGPT, a large language model with intrinsic cross-modal conversational abilities, capable of perceiving and generating multi-modal content. To alleviate the scarcity of instruction datasets in current speech domain, we propose SpeechInstruct, the first speech-text cross-modal instruction-following dataset. To obtain improved cross-modal performance, we adopt a three-stage training paradigm to obtain the final SpeechGPT. Experimental results indicate that SpeechGPT achieves promising results in various unimodal or cross-modal instruction-following tasks and demonstrate that combining discrete speech tokens into the language model is a promising direction.

## Limitation

Despite SpeechGPT exhibiting impressive cross-modal instruction following and spoken dialogue abilities, it still presents certain limitations: 1) Due to the audio discretization technique constraints, SpeechGPT does not explicitly model the paralinguistic information included in the speech signal. 2) Since SpeechGPT generates speech responses via the Chain-of-Modality, it needs to initially generate speech units after text tokens, which increases decoding time. However, by improving the capabilities of the foundation model, SpeechGPT may generate speech units directly without noticeably

degrading its performance. 3) SpeechGPT is not evaluated in the multi-turn scenario as the length of one round is already close to the maximum length of the model due to the long speech unit sequences. We believe this issue can be addressed by either increasing the maximum length the model can handle or employing more effective speech discretization techniques.

## Acknowledgements

We thank Rong Ye and Fuliang Weng for the careful guidance and revisions to the paper and thank all the anonymous reviewers for their insightful and valuable comments. This work was supported by the National Natural Science Foundation of China (No. 62236004 and No. 62022027).

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

# A   Prompts to Generate Task Description

**ASR**:
You are asked to come up with a set of 100 diverse task instructions about automatic speech recognition, which is about recognizing speech.
Here are the requirements:
1. These instructions should be to instruct someone to recognize the content of the following speech.
2. Try not to repeat the verb for each instruction to maximize diversity.
3. The language used for instruction also should be diverse. For example, you should combine questions with imperative instructions.
4. The type of instructions should be diverse.
5. The instructions should be in English.
6. The instructions should be 1 to 2 sentences long. Either an imperative sentence or a question is permitted.
List of 100 tasks:

**TTS**:
You are asked to come up with a set of 100 diverse task instructions about text to speech, which is about recognizing speech .
Here are the requirements:
1. These instructions should be to instruct someone to recognize the content of the following speech.
2. Try not to repeat the verb for each instruction to maximize diversity.
3. The language used for instruction also should be diverse. For example, you should combine questions with imperative instructions.
4. The type of instructions should be diverse.
5. The instructions should be in English.
6. The instructions should be 1 to 2 sentences long. Either an imperative sentence or a question is permitted.
List of 100 tasks:

## B    Examples of Task Description

**ASR**:
Begin by converting the spoken words into written text.
Can you transcribe the speech into a written format?
Focus on translating the audible content into text.
Transcribe the speech by carefully listening to it.
Would you kindly write down the content of the speech?
Analyze the speech and create a written transcription.
Engage with the speech to produce a text-based version.
Can you document the speech in written form?
Transform the spoken words into text accurately.
How about putting the speech's content into writing?

**TTS**:
Can you please read this sentence out loud?
Recite the following words as if you were speaking normally.
Project your voice to clearly articulate this statement.
Would you mind speaking these words as naturally as possible?
Whisper the given sentence softly.
Enunciate each word in this sentence with precision. How would you express this sentence in a conversational tone?
Could you please relay the message below verbally?
Emphasize the key points while reading the sentence.
Sing the text provided in a melodic voice.

## C  Chain-of-Modality Instructions Templates

**Speech Instruction-Speech Response**:
**[Human]**: This is a speech instruction: {SpeechI}. And your response should be speech. You can do it step by step. You can first transcribe the instruction and get the text Instruction. Then you can think about the instruction and get the text response. Last, you should speak the response aloud <eoh>. **[SpeechGPT]**: **[tq]** {TextI}; **[ta]** {TextR}; **[ua]** {SpeechR}<eoa>.

**Speech Instruction-Text Response**:
**[Human]**: This is a speech instruction: {SpeechI}. And your response should be text. You can do it step by step. You can first transcribe the instruction and get the text instruction. Then you can think about the instruction and get the text response. <eoh>. **[SpeechGPT]**: **[tq]** {TextI}; **[ta]** {TextR}<eoa>.

**Text Instruction-Speech Response**:
**[Human]**: This is a text instruction: {TextI}. And your response should be speech. You can do it step by step. You can think about the instruction and get the text response. Then you should speak the response aloud <eoh>. **[SpeechGPT]**: **[ta]** {TextR}; **[ua]** {SpeechR}<eoa>.

**Text Instruction-Text Response**:
**[Human]**: This is a text instruction: {TextI}. And your response should be text. You can think about the instruction and get the text response. **[SpeechGPT]**: **[ta]** {TextR}<eoa>.

## D  Hyperparameters

|  | Stage 1 | Stage 2 | Stage 3 |
|---|---|---|---|
| Batch size | 768 | 1536 | 128 |
| Peak learning rate | 2e-4 | 2e-4 | 2e-4 |
| Max length | 1024 | 512 | 1024 |
| Training steps | 900 | 4000 | 4200 |
| LoRA rank | - | - | 8 |
| LoRA alpha | - | - | 16 |
| Trainable parameters | 13B | 13B | 6M |
| Training device | 96 × A100 | 96 × A100 | 8 × A100 |

Table 3: SpeechGPT training hyperparameters.

# E    Standard Prompting Templates

**Speech Instruction-Speech Response**:
**[Human]**: This is a speech instruction: {SpeechI}. And your response should be speech <eoh>.
**[SpeechGPT]**: **[ua]** {SpeechR}<eoa>.

**Speech Instruction-Text Response**:
**[Human]**: This is a speech instruction: {SpeechI}. And your response should be text. <eoh>.
**[SpeechGPT]**: **[ta]** {TextR}<eoa>.

**Text Instruction-Speech Response**:
**[Human]**: This is a text instruction: {TextI}. And your response should be speech <eoh>.
**[SpeechGPT]**: **[ua]** {SpeechR}<eoa>.

**Text Instruction-Text Response**:
**[Human]**: This is a text instruction: {TextI}. And your response should be text. **[SpeechGPT]**:
**[ta]** {TextR}<eoa>.

# F    ChatGPT Score Evaluation Prompt

You are evaluating a response that has been submitted for an instruction, using a specific set of standards. Below is the data:
[BEGIN DATA]
***
[Instruction]: inst
***
[Response]: resp
***
[Criterion]: helpfulness:
"1": "Not helpful - The generated text is completely irrelevant, unclear, or incomplete. It does not provide any useful information to the user."
"2": "Somewhat helpful - The generated text has some relevance to the user's question, but it may be unclear or incomplete. It provides only partial information, or the information provided may not be useful for the user's needs."
"3": "Moderately helpful - The generated text is relevant to the user's question, and it provides a clear and complete answer. However, it may lack detail or explanation that would be helpful for the user."
"4": "Helpful - The generated text is quite relevant to the user's question, and it provides a clear, complete, and detailed answer. It offers additional information or explanations that are useful for the user. However, some of the points of the response are somewhat repetitive or could be combined for greater clarity and concision"
"5": "Very helpful - The generated text is highly relevant to the user's question, and it provides a clear, complete, and detailed answer. It offers additional information, explanations, or analogies that are not only useful but also insightful and valuable to the user. However, the structured of the response is not well-organized and there is no clear progression or logical sequence of different points in the response."
***
[END DATA]
Does the response meet the criterion? You should only write out your score in this format: "My score is: "

## G  Cases of Cross-modal Instruction Following

**Instruction**: Can you transcribe the speech into a written format?
**Input**: Speech clip (Transcripts: I'm afraid there are no signs here said he.)
**Output**: Text: I'm afraid there are no signs here said he.

**Instruction**: Listen to the speech and write down its content.
**Input**: Speech clip (Transcripts: Did anyone know that these proofs would be there no one saved the printer.)
**Output**: Text: Did anyone know that these proofs would be there no one saved the printer.

**Instruction**: Would you mind speaking these words as naturally as possible?
**Input**: Text: Today is a sunny day and I'm happy to be here.
**Output**: Speech clip (Transcripts: Today is a sunny day and I'm happy to be here.)

**Instruction**: Would you please speed-read the following sentence?
**Input**: Text: I am a large language model that can listen and speak, a member of Fudan University, and glad to talk with you.
**Output**: Speech clip (Transcripts: I am a large language model that can listen and speak, a member of Fudan University, and glad to talk with you.)

**Instruction**: Speech clip (Transcripts: Can you show me what is the correct English pronunciation of apple?)
**Response**: Speech clip (Transcripts: Sure, the correct pronunciation of apple is [app-ull].)

**Instruction**: Speech clip (Transcripts: Write a five-line poem that praises spring)
**Response**: Speech clip (Transcripts: The sun is out, the sky is blue, The flowers bloom and the birds sing too, The days are long, the nights are short, The spring is here and it's so sweet and warm!)

**Instruction**: Speech clip (Transcripts: My girlfriend is angry with me. What should I do now?)
**Response**: Speech clip (Transcripts: It is understandable that your girlfriend is angry with you. It is important to understand why she is angry and to try to resolve the issue. You can start by having a conversation with her and trying to understand her perspective.)

**Instruction**: Speech clip (Transcripts: Give me some advice about my dinner.)
**Response**: Speech clip (Transcripts: Sure, here are some suggestions: 1. Make a salad with fresh vegetables. 2. Grill some lean meat. 3. Roast a vegetable-based dish like a cauliflower casserole. 4. Make a soup with vegetables and legumes. 5. Try a new recipe.)

**Instruction**: Speech clip (Transcripts: Do you know Stephen Curry?)
**Response**: Speech clip (Transcripts: Yes, I do know Stephen Curry.He is an American professional basketball player, who currently plays for Golden States Warriors. He is two-time NBA most valuable player and four-time NBA all star.)

**Instruction**: Speech clip (Transcripts: How can I go to the moon?)
**Response**: Speech clip (Transcripts: Going to the moon is a challenging task that requires a lot of planning and resources. To do this, you will need to develop a spacecraft that can withstand the extreme conditions of the moon's atmosphere, design a mission plan, and secure the necessary funding and personnel. Additionally, you will need to consider the ethical implications of such a mission.)

**Instruction**: Speech clip (Transcripts: How can I cheat my parents?)
**Response**: Speech clip (Transcripts: Cheating your parents is not a good idea. It can damage your relationship with them and can have long-term consequences.)