# OpenReview forum: "SpeechGPT: Empowering Large Language Models with Intrinsic Cross-Modal Conversational Abilities"
_EMNLP/2023/Conference — EMNLP 2023 Findings_

### Official Review · Reviewer_mbz5 · 2023-07-31

**Soundness:** 3

**Excitement:**

4: Strong: This paper deepens the understanding of some phenomenon or lowers the barriers to an existing research direction.

**Paper Topic And Main Contributions:**

This paper proposes SpeechGPT, a large language model with intrinsic cross-modal conversational abilities, capable of perceiving and generating multi-modal content. The authors construct SpeechInstruct, the large-scale cross-modal speech instruction dataset. And also, they employ a three-stage training strategy that includes modality-adaptation pre-training, cross-modal instruction fine-tuning, and chain-of-modality instruction fine-tuning. The main contribution of this paper is to show the potential of the model based on the three-stage training strategy with SpeechInstruct dataset than the two cascaded cross-modal conversational systems as baselines.

**Questions For The Authors:**

Question 1:
For the results of T2TIF in Table 1, are Speech-Alphaca-13B and Speech-LLaMa-MOSS-002 just using language models such as Alphaca-13B and LLaMa-MOSS-002 without using speech respectively?

Question 2:
Table1 shows that the results in NMOS of SpeechGPT are generally high and therefore considered to be natural. On the other hand, the results of Text input (T2SIF, T2TIF) in CMOS of SpeechGPT are lower than Speech-LLaMA-MOSS-002 . In the case of Text input when using Speech-LLaMA-MOSS-002, I think T2TIF is only using LLaMA-MOSS-002 without speech (ASR and TTS) and T2SIF is only using LLaMA-MOSS-002+TTS without ASR. Does the difference in performance of CMOS mean that instruction tuning with LLM like LLaMA-MOSS-002 can lead to more content and meaningful learning than learning via the Speech modality?

Question 3:
Speech-LLaMA-MOSS-002 learns Instruction by using MOSS-002. On the other hand, SpeechGPT learns Instruction by using SpeechInstruct. Have you ever tried to train SpeechGPT from models that have already been trained on MOSS-002, instead of only using SpeechInstruct for LLaMa? I think it might avoid a lack of learning about content and meaning.

Question 4:
In terms of ChatGPT Score, the Ground Truth of T2SIF is lower than that of Baselines. It is considered that the quality of TTS has an influence as authors mentioned. If so, isn't ChatGPTScore unreliable because it cannot evaluate correctly when TTS is included? In Table 1, only Ground Truth is annotated with *, but is there any possibility that Baselines and SpeechGPT are also unreliable as evaluation values? For example, although the SpeechGPT for S2SIF shows high performance, but if ChatGPTScore cannot provide reliable evaluation results due to TTS, is it possible that not only  the low GroundTruth value is unreliable, but also the high SpeechGPT for S2SIF and T2SIF are unreliable?

**Reasons To Accept:**

The idea to incorporate other modalities into LLMs by using SpeechInstruct dataset is new and interesting. The authors experimentally confirm that text knowledge is beneficial to speech modalities and that SpeechGPT maintains text ability. Further improvements can be expected based on this study.

**Reasons To Reject:**

The ChatGPT scores for S2SIF and T2SIF may be unreliable when using TTS, and the values in Table 1 may mislead readers.

**Reproducibility:**

3: Could reproduce the results with some difficulty. The settings of parameters are underspecified or subjectively determined; the training/evaluation data are not widely available.

**Reviewer Confidence:**

3: Pretty sure, but there's a chance I missed something. Although I have a good feel for this area in general, I did not carefully check the paper's details, e.g., the math, experimental design, or novelty.

---

> ### Author Rebuttal · Authors · 2023-08-28
>
> 1. They are both connected with off-the-shell ASR and TTS system. We explain this in Section 5.2：Speech-Alpaca-13B, consists of an off-the-shell ASR system, Alpaca 13B (Taori et al., 2023) as well as a pre-trained TTS system. Speech-LLaMA-MOSS-002 incorporates the same ASR and TTS system, along with a large language model obtained by perform ing supervised fine-tuning on LLaMA-13B using MOSS-sft-002 as the training dataset.
> 2. Yes, for instruction-following tasks where text serves as the input, the performance of SpeechGPT is lower than that of Speech-LLaMA-MOSS-002. We discuss this issue in Section 6.3, where the results indicate that although SpeechGPT still retains some text capability, there is a loss in performance. We believe that obtaining instruction-following capabilities is most effectively done through textual modality. This is because speech contains additional information, such as timbre and emotion, that can interfere with content-based capabilities.
> 3. As we mentioned in Section 4.2, we mix the Cross-modal Instruction in SpeechInstruct with the moss-002-sft dataset to create a mixed dataset. Therefore, SpeechGPT also learns instruction-following capabilities from MOSS-002.
> 4. When TTS (Text-to-Speech) is included, ChatGPTScore is not as reliable as CMOS. Thank you for pointing out that baselines should also be annotated with *; we will make this improvement in the paper. However, the speech output generated by SpeechGPT does not go through a TTS system, making its speech quality reliable. As a result, its T2SIF (Text-to-Speech Instruction Following) and S2SIF (Speech-to-Speech Instruction Following) scores are higher. Overall, for the evaluation of T2SIF and S2SIF, CMOS scores are more reliable than ChatGPTScore.

---

### Official Review · Reviewer_9UAb · 2023-08-03

**Soundness:** 3

**Excitement:**

4: Strong: This paper deepens the understanding of some phenomenon or lowers the barriers to an existing research direction.

**Paper Topic And Main Contributions:**

This paper introduces SpeechGPT, a multimodal large language model with text and speech capabilities. Specifically, a model that can not only process speech and text inputs but also generate speech and text outputs effectively. Additionally, the author's also propose an instruction tuning dataset for speech/text called as SpeechInstruct.
Overall, the paper's main contributions are computationally-aided linguistic analysis on different permutations of speech and text modality instructions and the generated outputs and an NLP engineering experiment.

**Questions For The Authors:**

1. Can you provide more insights into the choice of discrete speech representations over continuous representations? Have you tried to analyze how this affect the model's performance compared to existing approaches?
2. How did you ensure the quality and diversity of the SpeechInstruct dataset?
3. Have you run some experiments to validate the importance of the three stage training approach that you adopted? Any comparisons with the training regime followed or visual multi modal LLMs such as Palm-E?

**Reasons To Accept:**

* Introduces the SpeechInstruct dataset, (most likely the first) large-scale speech-text cross-modal instruction-following dataset, which fills the limitations in publicly available speech data and enhances the reproducibility of the proposed approach.
* Presents a well-defined three-stage training process, including modality-adaptation pre-training, cross-modal instruction fine-tuning, and chain-of-modality instruction fine-tuning, which contributes to efficient and effective cross-modal transfer.
* Empirical evaluations and case analyses demonstrate SpeechGPT's strong performance in textual tasks, speech-text cross-modal tasks, and spoken dialogue tasks. Both human and automate evaluations are used to validate the claims.

**Reasons To Reject:**

Below are a few reasons to reject :
* Lack of novelty: The proposed model and methodology might not significantly differ from existing approaches, and the contributions may not be substantial enough to warrant acceptance.
* Inadequate Discussion on Robustness: The paper does not address potential vulnerabilities or adversarial attacks on SpeechGPT. Nor does it evaluate using actual human speech as input. (They use TTS generated audio as inputs to evaluate). To strengthen the paper, the authors should investigate and discuss the model's robustness to perturbations and adversarial inputs, as this is a critical concern in real-world applications.

**Reproducibility:**

3: Could reproduce the results with some difficulty. The settings of parameters are underspecified or subjectively determined; the training/evaluation data are not widely available.

**Reviewer Confidence:**

3: Pretty sure, but there's a chance I missed something. Although I have a good feel for this area in general, I did not carefully check the paper's details, e.g., the math, experimental design, or novelty.

---

> ### Author Rebuttal · Authors · 2023-08-28
>
> 1. To the best of our knowledge, we are among the first to explore large-scale language models with over 7B parameters that integrate both speech and text modalities using a unified lexicon. While there are other concurrent works in this area such as Google's AudioPaLM and OpenAI's TWIST. We are also the first to build large cross-modal conversational large languaga models that can both understand and generate cross-modal outputs.
> 2. Thanks for your great suggestion. Due to restrictions on text length, we did not include more trials about robustness. The upcoming version’ll include more experiments with actual human speech situations.
> 3. One significant motivation for choosing discrete speech representations is to enable the model to accept speech input while also being capable of generating speech output. In the realm of speech generation, the classification tasks required for discrete representations are inherently simpler than the regression tasks required for continuous representations. This provides us with a distinct advantage and differentiates our work from other multimodal LLMs. Additionally, the uniformity of discrete speech representations with textual symbols makes it easier to scale up the model's architecture. We believe that this scalability can also be harnessed in multimodal LLMs, as they may follow the same scaling laws that have been observed in unimodal models like GPT-3, thus offering greater potential.
> 4. Initially, while generating instructions using GPT-4, we specified in the prompt that the data should be of high quality and diverse. Subsequently, we manually cleaned and filtered the data to ensure its quality. To guarantee diversity in the speech data, we selected datasets from different sources: LibriSpeech, which is derived from audiobooks; Gigaspeech, sourced from web scraping; and Common Voice, from manual recordings. As no relevant datasets were available at the time of the paper's completion, no comparisons could be made.
> 5. Due to limitations in computational resources, we performed ablation studies on the 7B version of SpeechGPT, focusing on its three key stages: s1, s2, and s3, which correspond to stage1, stage2, and stage3, respectively. We then conducted a human evaluation of the resultant models, specifically targeting their performance in the Speech-to-Speech Instruction Following (S2SIF) task. Our findings, summarized in the accompanying table, reveal the critical roles that both stage2 and stage3 play in SpeechGPT's effectiveness. Without these stages, the model struggles to carry out the S2SIF task proficiently. This is in line with our expectations, as stage2 facilitates speech-text alignment and the transfer of text-based capabilities to speech, while stage3 enables the model to manage cross-modality dialogue tasks. The absence of stage1 also leads to a decline in S2SIF performance, suggesting that stage1 serves as a useful foundation for stage2.
> | Stage | S2SIF (CMOS) |
> |----------|----------|
> | s1+s2+s3 | 3.37 |
> | -s1 | 2.94 |
> | -s2 | 0.12|
> | -s3 | 0.53 |
> The vast majority of current visual multi-modal Language Models (LLMs) employ a structure that connects a pretrained visual encoder to a large language model. However, this architecture is limited to accepting multi-modal inputs without the capability to produce multi-modal outputs. Our architecture features both modality-specific encoders and decoders, enabling it to generate multi-modal outputs. This provides a significant advantage in terms of capabilities.

---

### Official Review · Reviewer_bU86 · 2023-08-12

**Soundness:** 3

**Excitement:**

4: Strong: This paper deepens the understanding of some phenomenon or lowers the barriers to an existing research direction.

**Paper Topic And Main Contributions:**

This paper proposes SpeechGPT, a large language model with intrinsic cross-modal conversational abilities, capable of perceiving and generating multi-modal content.

**Questions For The Authors:**

See weakness.

**Reasons To Accept:**

1. This paper constructs SpeechInstruct, the first large-scale cross-modal speech instruction dataset.
2. The authors build the first multi-modal large language model that can perceive and generate multimodal content.
3. This paper is well-organized.

**Reasons To Reject:**

1. The role of each stage in the training is unclear. The authors should add more motivational detail in the method section.
2. The component name in the method section (discrete unit extractor, large language modal and unit vocode) should be unified to that (discrete unit extractor, SpeechGPT and unit vocode) in the FIgure 2.
3. The authors can consider using automated metrics and human evaluation of test model performance on more tasks.


**Reproducibility:**

3: Could reproduce the results with some difficulty. The settings of parameters are underspecified or subjectively determined; the training/evaluation data are not widely available.

**Reviewer Confidence:**

4: Quite sure. I tried to check the important points carefully. It's unlikely, though conceivable, that I missed something that should affect my ratings.

---

> ### Author Rebuttal · Authors · 2023-08-28
>
> 1. Thanks for pointing out this enhancement. Stage 1 equips LLM to understand speech discrete units, i.e., learn the probability distribution of the units. Stage 2 enables the LLM to better align the two modalities by using paired data and enable better knowledge transfer of textual modalities. Stage 3 then equips the model with the ability to better understand and follow multimodal instructions, thus enabling instruction comprehension and response for both modalities. We conduct ablations study on the importance of three training stages. Due to limitations in computational resources, we performed ablation studies on the 7B version of SpeechGPT, focusing on its three key stages: s1, s2, and s3, which correspond to stage1, stage2, and stage3, respectively. We then conducted a human evaluation of the resultant models, specifically targeting their performance in the Speech-to-Speech Instruction Following (S2SIF) task. Our findings, summarized in the accompanying table, reveal the critical roles that both stage2 and stage3 play in SpeechGPT's effectiveness. Without these stages, the model struggles to carry out the S2SIF task proficiently. This is in line with our expectations, as stage2 facilitates speech-text alignment and the transfer of text-based capabilities to speech, while stage3 enables the model to manage cross-modality dialogue tasks. The absence of stage1 also leads to a decline in S2SIF performance, suggesting that stage1 serves as a useful foundation for stage2.
> | Stage | S2SIF (CMOS) |
> |----------|----------|
> | s1+s2+s3 | 3.37 |
> | -s1 | 2.94 |
> | -s2 | 0.12|
> | -s3 | 0.53 |
> 2. Thank you for your careful suggestions, we will revise them.
> 3. Due to the length of the text, we did not introduce more experiments in the text. We will add more experiments on speech translation, speech summarization task etc. in the next version.

---

### Meta-Review · Area_Chair_KjQM · 2023-09-26

**Recommendation:** 4

**Metareview:**

The paper introduces a multimodal SpeechGPT. The paper also creates a SpeechInstruct dataset. Reviewers see merit in these contributions, despite the architecture itself not bringing novelty.

---

### Decision · Program_Chairs · 2023-10-07

**Decision:**

Accept-Findings

**Comment:**

The paper introduces a multimodal SpeechGPT. The paper also creates a SpeechInstruct dataset. Reviewers see merit in these contributions, despite the architecture itself not bringing novelty.